# Sub-cone visual resolution by active, adaptive sampling in the human foveola

Jenny L Witten*, Veronika Lukyanova, Wolf M Harmening

Department of Ophthalmology, Rheinische Friedrich-Wilhelms-Universität Bonn, Bonn, Germany

## eLife Assessment

This **important** work uses in vivo foveal cone-resolved imaging and simultaneous microscopic photo-stimulation to investigate the relationship between ocular drift - eye movements long thought to be random - and visual acuity. The surprising result is that ocular drift is systematic - causing the object to move to the center of the cone mosaic over the course of each perceptual trial. The tools used to reach this conclusion are state-of-the-art and the evidence presented is **convincing**. This work advances our understanding of the visuomotor system and the interplay of anatomy, oculomotor behavior, and visual acuity.

## Abstract

The foveated architecture of the human retina and the eye's mobility enables prime spatial vision, yet the interplay between photoreceptor cell topography and the constant motion of the eye during fixation remains unexplored. With in vivo foveal cone-resolved imaging and simultaneous microscopic photo stimulation, we examined visual acuity in both eyes of 16 participants while precisely recording the stimulus path on the retina. We find that resolution thresholds were correlated with the individual retina's sampling capacity, and exceeded what static sampling limits would predict by 18%, on average. The length and direction of fixational drift motion, previously thought to be primarily random, played a key role in achieving this sub-cone diameter resolution. The oculomotor system finely adjusts drift behavior towards retinal areas with higher cone densities within only a few hundred milliseconds to enhance retinal sampling.

**\*For correspondence:**
jennylwitten@gmail.com

**Competing interest:** The authors declare that no competing interests exist.

## Introduction

Assessing visual abilities was already important in historic times (*Bohigian, 2008*), and the precise measurement of visual acuity, our ability to resolve fine spatial detail by eye, has great importance for many real-life scenarios and is up to this day the primary diagnostic tool to determine visual function in a clinical and optometric setting. Quite surprisingly, the widely-believed assumption that the packing density and arrangement of retinal photoreceptors at the foveal center set the limit to this ability has never been experimentally confirmed.

Fovealization, the morphological and functional specialization of the cellular architecture of the light-sensitive retina optimizes the human eye for high-acuity daytime vision (*Caves et al., 2018*; *Tuten and Harmening, 2021*). Within the central one-degree diameter of the fovea, termed foveola, postreceptoral neurons are displaced centrifugally and the area is free of potentially shadowing blood vessels and glia cells (*Hendrickson and Yuodelis, 1984*; *Syrbe et al., 2018*). The outer segments of foveolar cone photoreceptors are maximally thinned and densely packed for peak spatial sampling (*Hirsch and Curcio, 1989*; *Rossi and Roorda, 2010*; *Williams and Coletta, 1987*), which at the same time makes these cells the most difficult to study ex vivo (*Curcio et al., 1987*) as well as in vivo (*Rossi et al., 2011*). Each foveolar cone synapses to one ON- and one OFF-midget bipolar cell, which in turn

synapse exclusively upon single ON- and OFF-midget ganglion cells, a circuit that is adult-like before birth (*Zhang et al., 2020*). This establishes an undisturbed *private line* from individual foveal receptors to central processing stages.

Based on indirect comparisons between histological and psychophysical data, the hypothesis that cone spacing imposes the fundamental limit for visual resolution has been put forward (*Curcio et al., 1990*; *Rossi and Roorda, 2010*). It is well established that cone spacing, especially in the central fovea, is highly variable between individuals (*Cava et al., 2020*; *Curcio et al., 1990*; *Reiniger et al., 2021*; *Wang et al., 2019*), making general comparisons between acuity measurements and foveolar density estimated from histological samples susceptible to error. One of the main reasons why the hypothesis lacks direct experimental proof is that because, under natural viewing conditions, both visual resolution and experimental access to foveal photoreceptors are blurred by the imperfect optics of the human eye (*Campbell and Green, 1965*; *Marcos et al., 2008*). Here, we have overcome the optical barrier of the human eye by employing adaptive optics cell-resolved in vivo retinal imaging in conjunction with micro-psychophysics to study directly whether the individual's mosaic of foveolar cones determines visual performance in a high-acuity resolution task.

While acuity is assumed to be mainly limited by the resolving capacity of the eye's optics and retinal mosaic, it is well established that, for different visual tasks, performance thresholds can be substantially lower than the sampling grain of photoreceptors. This phenomenon has been termed hyperacuity (*Westheimer, 1975*) and depends on the neural visual system's ability to extract subtle differences within the spatial patterns of the optical image on the retina (*Westheimer, 2012*). Thus, the visual system already incorporates mechanisms to detect relative spatial offsets an order of magnitude smaller than the spatial granularity of the retina. To make use of those fine distinctions in a resolution task, the neuronal system needs to go beyond purely spatial coding of incoming signals.

Unlike a camera, the visual system depends on temporal transients arising in the receptor's cellular signals. Neurons in the retina, thalamus, and later stages of the visual pathways respond strongly to temporal changes (*Kaplan and Benardete, 2001*; *Nagano, 1980*). Thus, the fovealized retinal architecture in humans is accompanied by a dynamic sampling behavior that, by quick and precise movements of the eye, brings retinal images of objects of interest to land in the foveola (*Ko et al., 2010*). Even during steady fixation, for example of a distant face or a single letter of this text, incessant fixational eye movements slide tens to hundreds of foveolar photoreceptors across the retinal image, thereby introducing temporal modulations that translate spatial activation patterns into the temporal domain (*Kuang et al., 2012*). Small and rapid gaze shifts known as microsaccades relocate the gaze within the foveola during periods of fixation (*Ko et al., 2010*), and between microsaccades, the eyes perform a more continuous, seemingly random motion termed fixational drift (*Intoy and Rucci, 2020*; *Krauskopf et al., 1960*). Computational work suggested that fixational eye motion would introduce noise and thus impair visual acuity (*Burak et al., 2010*; *Pitkow et al., 2007*). Contrarily, recent studies on human psychophysics demonstrated fixational eye motion to be beneficial for fine spatial vision (*Intoy and Rucci, 2020*; *Rolfs, 2009*; *Rucci et al., 2007*). Especially drift motion has been increasingly argued to not just be randomly refreshing neural activity, but rather structuring it (*Clark et al., 2022*; *Hafed et al., 2021*; *Intoy and Rucci, 2020*) and being under central control (*Herrmann et al., 2017*).

The incessant motion of the eye conveys fine spatiotemporal detail that requires deciphering of continuously changing photoreceptor signals, which are linked by the geometry of the photoreceptor array and by how the eye moves. For instance, luminance modulation in individual cones will scale with drift length. Larger luminance variations on single receptors also yields more neuronal activity within the range of temporal frequencies parvocellular ganglion cells are sensitive to. Selective spatial frequencies can thus be amplified by varying drift lengths (*Intoy and Rucci, 2020*). While the neuronal mechanisms that generate fixational drift are still not fully understood (*Ben-Shushan et al., 2022*), its consequence to visual perception has been demonstrated. Drift was shown to improve visual performance in resolution tasks (*Intoy and Rucci, 2020*; *Ratnam et al., 2017*; *Rucci et al., 2007*). Indeed, considerable differences in ocular drift between individuals exist (*Cherici et al., 2012*; *Clark et al., 2022*), and subjects exhibiting less drift were shown to have better acuity (*Clark et al., 2022*). If such differences are a consequence of an active, adaptive mechanism, however, and how drift behavior is related to the photoreceptors that sample the retinal image is unknown.

The direct experimental access to the foveolar center, when other limiting factors like image blur or retinal motion are taken out of the equation or can be precisely measured, will allow to confirm or

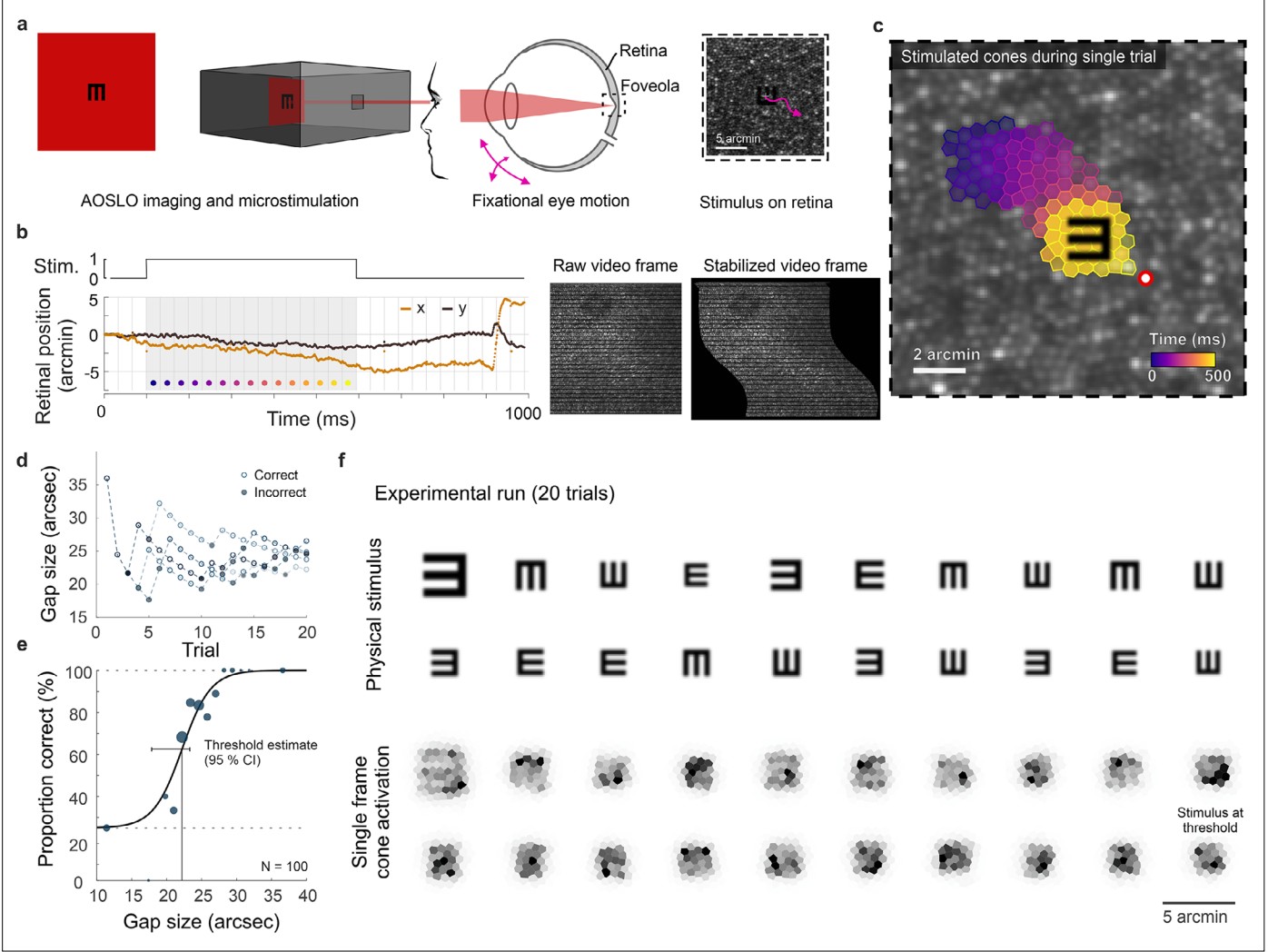

**Figure 1.** Cone-resolved adaptive optics micro-psychophysics. (**a**) Schematic of cell-resolved visual acuity testing in the human foveola with an adaptive optics scanning laser ophthalmoscope (AOSLO). Stimuli were dark Snellen-E optotypes presented at variable sizes and four orientations in the center of the 788 nm AOSLO imaging raster. Participants responded by indicating stimulus orientation during natural viewing, i.e., unrestricted eye motion. (**b**) Exemplary single trial retinal motion trace and strip-wise image stabilization of a single AOSLO frame (shown here during a microsaccade for better visibility). Trials containing microsaccades or blinks during the 500 ms stimulus presentation (gray shaded area) were excluded. The x-axis grid represents individual video frames (33 ms). (**c**) Foveolar retinal cone mosaic with a exemplary single trial retinal motion across the stimulus. Time is represented by color from stimulus onset to offset (purple to yellow). The cone density centroid (CDC) is shown as a red circle with white fill. (**d**) Typical psychophysical data of five consecutive runs in one eye. Each run followed a QUEST procedure with 20 trials. (**e**) Psychometric function fit to the data (about 100 trials). Acuity thresholds were estimated at 62.5% correct responses. (**f**) Exemplary retinal images (upper rows) and corresponding cone activation patterns (lower rows) of one experimental run (20 trials from top left to bottom right). Cone activation patterns are shown for a representative single frame. See *Videos 1 and 2* for a real-time video representation.

reject the long-standing hypothesis about the individual limits of vision. This will help to understand the fundamental physiological limitations of the visual system and will have important implications for clinical studies of retinal health.

## Results

### Resolution is finer than single cone sampling limits

We investigated the limitations of the photoreceptor packing density on individual visual resolution acuity by overcoming the optical aberrations of the eye with adaptive optics scanning laser ophthalmoscopy (AOSLO), while simultaneously performing psychophysical measurements and recording the

fixational retinal motion (*Figure 1a, b and c*). In a four-alternative forced-choice task, 16 healthy participants indicated the orientation of an E-optotype while inspecting the stimulus with their individually preferred fraction of foveolar photoreceptors. These cone photoreceptors were simultaneously imaged and it was later identified which cells contributed to resolving the stimulus (*Figure 1c and f*). A psychometric fit to the data expressed as percentage correct from 100 trials was used to compute visual acuity thresholds (see online Methods and *Figure 1d and e*). In this near diffraction-limited testing condition, participants reached visual acuity thresholds between 20.6 and 28.5 arcsec (mean ± SD: 24.1±2.4 arcsec), which compares to 20/8 vision (logMAR = –0.4). All participants reached thresholds better than 20/10 vision (logMAR = –0.3), the last line of a typical clinical Snellen chart or projectors of acuity optotypes that are used in clinical as well as optometric daily routine.

Cone densities at the cone density centroid (CDC) ranged between 10,692 and 16,997 cones/deg$^2$, with an average density of 13,640 cones/deg$^2$ (Average peak cone density, PCD: 13,944 cones/deg$^2$, range: 10,823–17,309 cones/deg$^2$), comparable to previous reports (*Cava et al., 2020*; *Putnam et al., 2005*; *Wang et al., 2019*; *Wells-Gray et al., 2016*; *Wilk et al., 2017*; *Zhang et al., 2015*). The median sampling cone density ranged between 10,297 and 16,104 cones/deg$^2$ (mean: 13,149 cones/deg$^2$). Two experimental runs of the eyes with the highest and lowest sampling density are exemplarily shown in *Videos 1 and 2*. The two foveolar cone mosaic images were also visualized and overlayed with a Snellen E stimulus at average threshold size (*Figure 2a*). A static, theoretical prediction given by the Nyquist sampling limit would assume the high-density retina where each single cone diameter is smaller than the Snellen E's gap or bar is able to resolve the stimulus, whereas the low-density retina fails in identifying the correct orientation (schematic representation in *Figure 2b*). However, for our 788 nm testing condition, all participants reached individual resolution thresholds well below their Nyquist limit predicted by the spacing between rows of cones (*Figure 2c and d*). On average, visual acuity thresholds exceeded this theoretical prediction by 20% and 16% in dominant and non-dominant eyes, respectively. When participants performed the same resolution task with a longer infrared wavelength (840 nm) imaging background, the absolute thresholds were slightly higher and thus closer to the Nyquist limit. Visual acuity thresholds were on average 7% below and 2% above the Nyquist limit for dominant and non-dominant eyes, respectively. These absolute visual acuity thresholds were the only case where noteworthy differences arose between the 788 nm and 840 nm experimental conditions. For all other analyses, we found qualitatively similar results for either wavelength and therefore only report the 788 nm results throughout the manuscript.

For the first time, we could measure the direct relation between the individual foveolar cone photoreceptor sampling density and participant's visual resolution thresholds. We found the diffraction-limited visual acuity thresholds to be strongly correlated to the foveolar sampling density in dominant as well as fellow eyes (*Figure 2d*). The higher the cone density, the smaller the visual stimulus that could be resolved. The degree of correlation slightly differed for dominant (r$^2$=0.45, p=0.005) and non-dominant eyes (r$^2$=0.28, p=0.036), suggesting that up to 45% of the variance in inter-subject visual acuity can be explained by the individual cone sampling densities. Overall, participants reached significantly lower thresholds with their dominant eyes (average: 1.5 arcsec, SD ±1.1; paired t-test, p<0.001). Nevertheless, visual acuity thresholds were strongly correlated between dominant and

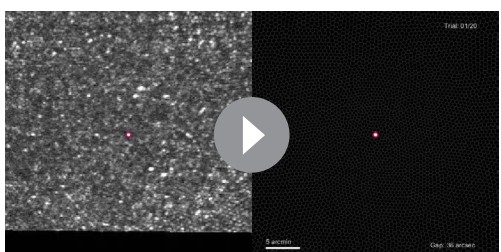

**Video 1.** Video recordings of one experimental run in the eye with highest sampling density. All successive single trial videos (left) and the Voronoi mosaic of cells colored with their respective amount of cone activation (right).

https://elifesciences.org/articles/98648/figures#video1

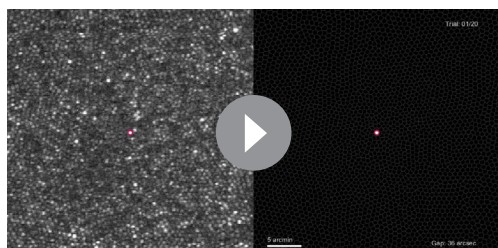

**Video 2.** Video recordings of one experimental run in the eye with lowest sampling density. All successive single trial videos (left) and the Voronoi mosaic of cells colored with their respective amount of cone activation (right).

https://elifesciences.org/articles/98648/figures#video2

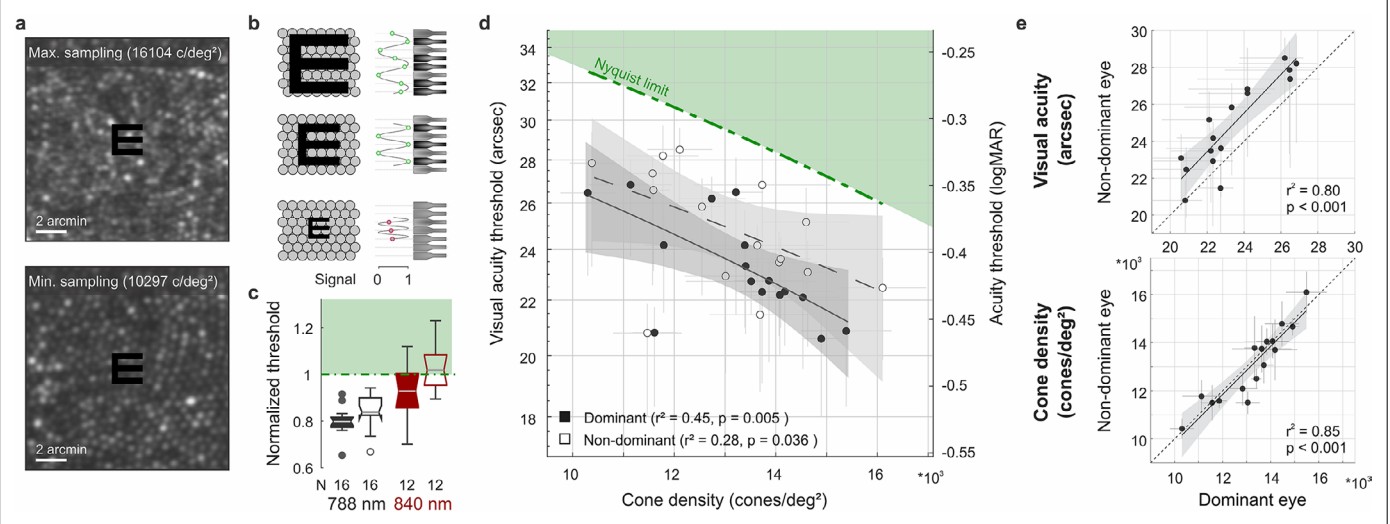

**Figure 2.** Visual acuity depends on foveolar sampling capacity. (**a**) Foveolar cone mosaics of the two eyes with highest and lowest cone densities, overlayed with the physical stimulus at an average threshold size (24 arcsec). (**b**) Nyquist limit: critical details equaling or larger than the spacing of cones are resolvable. (**c**) Visual acuity thresholds measured with 788 or 840 nm infrared light, normalized to the eyes' Nyquist limits. (**d**) Correlation between participant's individual visual acuity thresholds and cone density. Thresholds exceeded the Nyquist sampling limit and were significantly lower in eyes with higher cone densities. Dominant eyes are shown as filled, and non-dominant eyes as open markers. The gray horizontal and vertical bars at each point represent standard deviations of sampling cone density and the 95% confidence intervals for acuity thresholds. The theoretical Nyquist limit is represented by a dashed green line. (**e**) Correlation between dominant and non-dominant eyes in visual acuity (top) and cone density (bottom). Dominant eyes reached, on average, 1.5 arcmin lower thresholds than non-dominant eyes, whereas cone density (at the retinal locations that sampled the stimulus) was very similar between fellow eyes.

non-dominant eyes ($r^2$=0.80, p<0.001, *Figure 2e*). To test whether the effect of different absolute thresholds might be explained by underlying differences in the sampling cone density, fellow eye densities were compared to each other. Sampling densities had a very strong correlation between fellow eyes ($r^2$=0.85, p<0.001, *Figure 2e*), but did not differ between right and left eyes (p=0.38) nor when grouping them according to ocular dominance (p=0.88). This compares well with previous studies that also showed strong correlations between fellow eyes regarding both anatomical (*Cava et al., 2020*) as well as functional (*Reiniger et al., 2021*) characteristics. Dominant eyes had a median of 78 cones/deg² higher densities compared to their fellow eyes. To account for the 1.5 arcsec difference in acuity thresholds, a much higher density difference of about 1500 cones/deg² would have been needed. Based on these results, we conclude that the spatial arrangement of foveal cones can only partially predict resolution acuity. In the following, we show that ocular motion and its associated temporal modulations also influence visual resolution.

## Ocular drift is an active sampling mechanism

As the eye drifts, a visual stimulus projected onto the retina is processed as a spatiotemporal luminance flow. The stimulus itself as well as the extent of drift motion determine the characteristics of modulation. By analyzing the exact retinal locations sampling the stimulus we show the impact of the traveled path length first (*Figure 3*), followed by the direction of drift motion and its relation to anatomical and functional landmarks (*Figure 4*). In our experiments, we revealed that participants kept coming back to the same few hundreds of cone photoreceptors (*Figure 3a* and *Figure 3—figure supplement 1*). To focus on the characteristics and implications of drift eye motion, trials containing microsaccades during stimulus presentation were excluded from the analyses. During the short stimulus duration, however, microsaccades rarely occurred, as participants tend to suppress their microsaccades, likely because they can be detrimental to fine-scale discrimination (*Bowers et al., 2021*; *Intoy et al., 2021*). Drift motion patterns varied greatly across, but also within participants. Examples of drift motion paths for the eyes that performed the smallest and largest drift motion, on average, show a great variability in shapes as well as extent of motion (*Figure 3b*). In our analyses, we chose the drift length (sum of piecewise vector lengths) as the prime metric to describe the ocular drift motion,

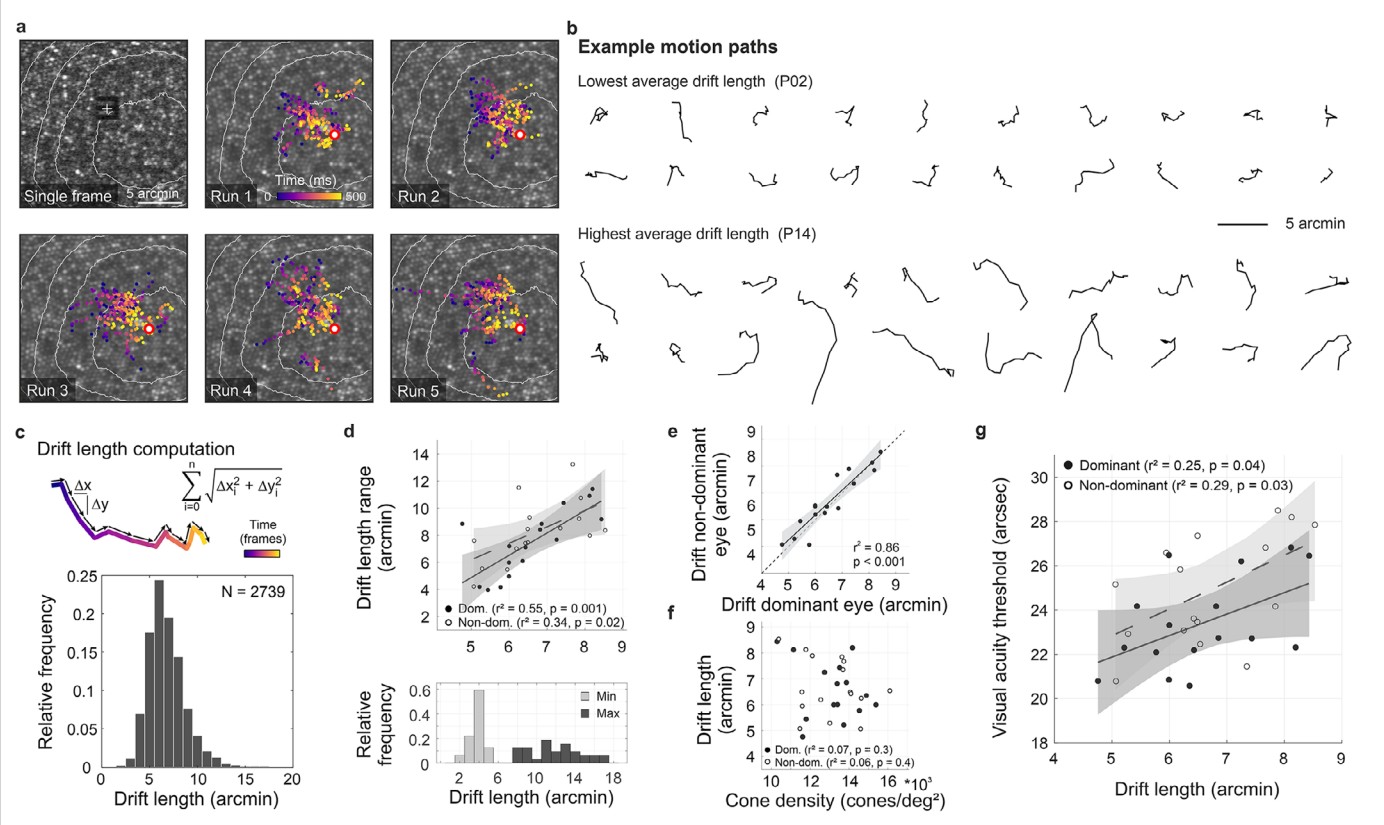

**Figure 3.** Fixational drift and the contribution to visual acuity. (**a**) Ocular drift during stimulus presentation (participant 16, left eye). Single adaptive optics scanning laser ophthalmoscopy (AOSLO) frame captured during Snellen E presentation (top left) and all single stimulus positions (colored dots) of five experimental runs shown on the corresponding cone mosaic (panels 2–6). White iso-lines delimit cone density percentile areas (90[th] to 50[th] percentile visible). Time is represented by color from stimulus onset to offset (purple to yellow). Individual drift trajectories for all eyes are shown in *Figure 3—figure supplement 1*. (**b**) Individual motion traces highlighting intra- and inter-subject drift variability. Traces are from one run in the participant with the lowest (upper rows) and highest (lower rows) average drift lengths. (**c**) Computation of drift length as a sum of interframe motion vectors (top) and the relative frequency of occurrences among all participants and trials (bottom). (**d**) Median drift length and drift length range showed a moderate correlation in dominant as well as non-dominant eyes (top). The minimum drift length was similar between participants (3.8±0.8 arcmin) whereas the maximum length varied about three times as much (12.0±2.7 arcmin). (**e**) Drift lengths in fellow eyes had a very strong correlation. (**f**) Cone density and drift length did not show a significant correlation in dominant or non-dominant eyes. (**g**) The median drift length had a moderate correlation with visual acuity threshold in dominant as well as non-dominant eyes. Dominant eyes are indicated by filled, non-dominant eyes by open markers.

The online version of this article includes the following figure supplement(s) for figure 3:

**Figure supplement 1.** Drift trajectories on foveolar mosaics.

because the randomness underlying alternative metrics of drift eye movements becomes increasingly questionable (see also Discussion). Across all participants and experimental trials, drift lengths ranged between 2.5 and 17.2 arcmin, with a median length of 6.5 arcmin (which corresponds to a velocity of 5–34.5 arcmin/s, median: 13 arcmin/s, *Figure 3c*). The drift lengths are slightly smaller than in previous non-AO studies, which is attributable to the viewing situation. The participants were looking at a very small imaging field within a completely dark periphery without distracting structures or stimuli. The smallest drift movement performed was similar among eyes (range: 2.5–5.4 arcmin), whereas the largest individual drifts differed more than three times as much (range: 7.7–17.2 arcmin). Therefore, the individual drift span was rather driven by the larger drift lengths of an eye and there was a strong correlation between median drift length and drift range (dominant eyes: $r^2$=0.55, p=0.002, non-dominant eyes: $r^2$=0.34, p=0.02, *Figure 3d*).

Between fellow eyes, which were measured consecutively, drift lengths had a very strong correlation ($r^2$=0.86, p<0.001, *Figure 3e*) with no significant difference between eyes (paired t-test, p=0.2). The median drift lengths of all eyes varied between 4.8 and 8.5 arcmin (mean ± SD: 6.6±1.1 arcmin). Individual visual acuity thresholds were significantly correlated with drift lengths (dominant: $r^2$=0.25,

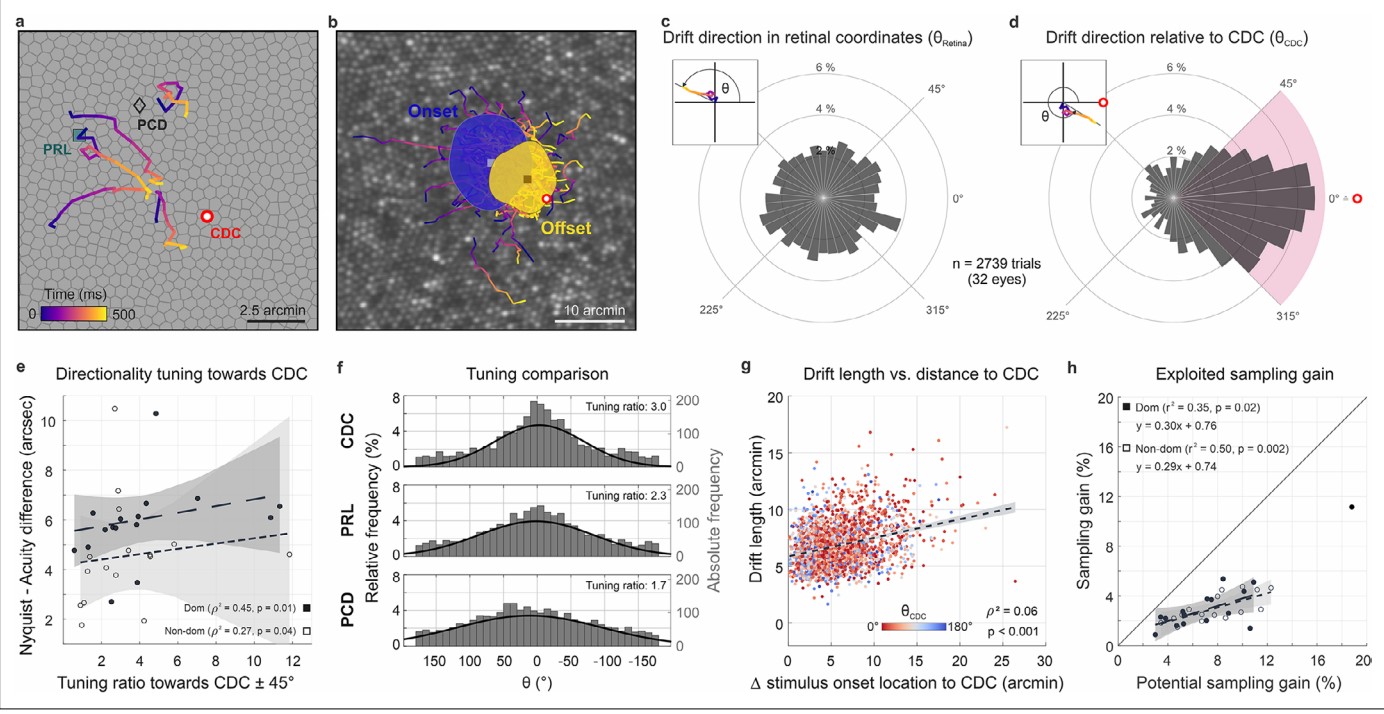

**Figure 4.** Drift moves stimuli to higher cone density areas. (**a**) Five exemplary motion traces relative to cone density centroid (CDC), preferred retinal locus (PRL), and peak cone density (PCD) location on the Voronoi tessellated cone mosaic of one participant. (**b**) All single trial motion traces of one eye are shown on the corresponding cone mosaic (95 trials containing drift only). One-SD isoline areas (ISOA) are shown for all stimulus onset (blue) and offset (yellow) locations, indicating a trend of directional drift towards higher cone densities during 500 ms stimulus presentation. (**c**) Polar histogram of all individual motion traces (n=2739) shows the relative frequency of motion angles, $\theta_{Retina}$, between the start (coordinate center) and end of motion in retinal coordinates. The inset indicates $\theta$ sign. (**d**) Same data as in c, where $\theta_{CDC}$ was computed relative to the line connecting drift start location and CDC, see inset. The pink quarter indicates the angular space used for the computation of the tuning ratio. For more details on the drift directionality of individual eyes, see *Figure 1*. (**e**) The difference between the acuity threshold and Nyquist limit showed a significant trend to be larger for stronger directionality tuning. The tuning ratio was computed as the ratio between the relative frequency of intra-participant drift motion towards the CDC (±45 deg) and the average of drift motion towards the remaining three quadrants. (**f**) Relative frequency of drift direction relative to CDC (top), PRL (middle), and PCD (bottom), respectively. For more details on the temporal progression of drift directionality, see *Figure 2*. (**g**) Across all participants and trials, drift length correlated with stimulus onset distance from CDC. There was no clear effect of stimulus onset distance on motion directionality (data color corresponding to $\theta_{CDC}$). (**h**) The achieved sampling gain due to the performed drift motion is significantly correlated to the potential sampling gain in individuals. In both dominant and non-dominant eyes the potential sampling gain is on average exploited by 30%, respectively. Due to shifting the CDC towards the stimulus, participants had different PRLs for a sustained fixation task and the visual resolution task (see *Figure 3*).

The online version of this article includes the following figure supplement(s) for figure 4:

**Figure supplement 1.** Individual drift directionality.

**Figure supplement 2.** Time course of drift directionality.

**Figure supplement 3.** Different retinal locations used in a fixation or resolution task.

p=0.04; non-dominant: r²=0.29, p=0.03, *Figure 3g*), with a trend towards better visual acuity for small ocular drift motion. On a photoreceptor resolved scale, this confirms recent findings which showed individual acuity thresholds to be correlated with the drift motion during a non-AO acuity task, closely related to the drift measured in a sustained fixation task (*Clark et al., 2022*).

Considering the previously shown correlation between visual acuity and sampling cone density, one could assume those two aspects to go along with an increase of ocular drift for lower cone densities, whereas higher densities potentially need less drift to translate the stimulus over the same number of cones. However, we don't find the drift motion to be tuned in a way to always let the stimulus slip across a similar number of cones. There was no significant correlation between cone densities and drift length (*Figure 3f*, dominant: r²=0.07, p=0.3; non-dominant: r²=0.06, p=0.4). Also, we do observe similar drift lengths across stimulus sizes. We note, however, that in all our experimental trials, stimulus sizes were quite similar. If drift length tuning existed, it may have been more pronounced with

a larger dispersion of stimulus sizes. In the following, we show that drift direction is indeed tuned to optimize sampling.

## Drift is adaptive and directed

Ocular drift has long been assumed to be a persistent jittery motion that follows random trajectories. Recent work showed that the amount of drift can vary and may be adapted to the task that has to be performed (*Clark et al., 2022*; *Intoy and Rucci, 2020*). We here investigated if, beyond this, humans are able to actively tune their ocular drift direction to exploit their prime spatial retinal processing properties. We, therefore, registered the individual drift motion trajectories with the photoreceptor mosaic, tracked them from the retinal location where the stimulus turned on (onset) to where it turned off after 500 ms (offset), and related these trajectories to foveolar landmarks (*Figure 4a and b*). Because of the individual retinal locations used for fixation before stimulus onset, we registered that, across all eyes, drift motion occurred towards all directions during stimulus inspection, and no general trend in drift eye movements towards a particular cardinal direction across participants occurred (*Figure 4c*). Individual eyes, however, showed different drift behavior mostly directed toward one or two of the four quadrants. All four cardinal directions were represented. Participant P8$_{right}$, for example, drifted towards the nasal or superior fovea in 90% of all trials. P14$_{right}$, on the other hand, drifted towards the temporal fovea in 75% of all trials.

When the frame of reference was rotated in each trial to register the motion from the onset location relative to the CDC, we found a clear directional bias in which the drift was likely to move the stimulus closer to the CDC. The drift directionality was evaluated by measuring the relative angle between drift onset to drift offset and drift onset to CDC. We observed a strong trend of drift directionality; 49% of all drift episodes moved the stimulus towards the CDC ± 45° (*Figure 4d*). The directionality was not pronounced directly after stimulus onset but increased with presentation duration (Rayleigh test for circular non-uniformity, p<0.001 for all conditions, see *Figure 4—figure supplement 2*). Among eyes, the individual fractions ranged between 16 and 80% of trials. Only two eyes drifted towards the CDC less frequently than given by chance (*Figure 4—figure supplement 1*). We computed the directionality tuning as the ratio of relative drift towards the CDC ± 45° (purple quadrant in *Figure 4d*) and the mean relative drift towards the three other quadrants. A ratio of 1 indicated the same relative frequency of drift towards all cardinal directions, whereas for a tuning ratio of 2 the retina moved the CDC towards the stimulus twice as often compared to each of the other three cardinal directions. The directionality tuning ratios ranged between 0.6 and 11.8 with a median value of 3. Directionality tuning ratios had a significant effect on how much the resolution threshold exceeded the Nyquist limit. Participants with highly tuned drift reached larger differences between the Nyquist limit and their visual acuity threshold (dominant eyes: $r^2$=0.45, p=0.01; non-dominant eyes: $r^2$=0.27, p=0.04, *Figure 4e*). Drift directionality was mostly similar between eyes, and if intra-ocular differences occurred, they were not related to ocular dominance. Also, we did not observe an effect of training on drift directionality: one of the two trained observers had a very strong drift directionality (7 and 11.8 in the dominant and non-dominant eye, respectively) while the other one exhibited a tuning ratio below average (2.1 and 2.3 in the dominant and non-dominant eye, respectively).

Next to the CDC, two other foveolar landmarks are often reported as anchor locations describing the center of the fovea. We here show that the CDC has the strongest relevance with respect to drift tuning. When relating the drift trajectories to the preferred retinal locus of fixation (PRL) or the location of peak cone density (PCD), we found a weaker approximation towards both. The retinae moved the stimulus towards the PRL or PCD

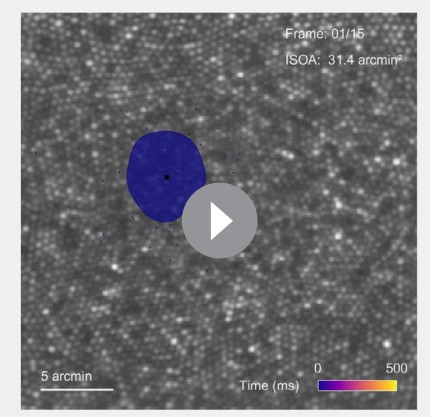

**Video 3.** Decrease of ISOA across trials during the 500 ms stimulus presentation. The area of stimulus onsets between trials shows a stronger variation than the area of stimulus offsets.

https://elifesciences.org/articles/98648/figures#video3

location in 42% or 35% of all trials, respectively (*Figure 4f*). Therefore, the observed directionality was strongest towards the CDC. In a considerable number of trials, the stimulus onset was further displaced from all of the three retinal locations and, therefore, a directed drift motion resulted in an approximation towards CDC as well as PRL and PCD. Also, in some eyes, two or all of these retinal locations lay very close together, which results in very similar effects. Nevertheless, in some eyes with particularly stable fixation that had at least a few arcmin distances between their PRL and CDC we repeatedly observed a stimulus onset close to PRL followed by a directional drift towards CDC with a resulting stimulus offset closer to the CDC. Across participants, this also resulted in a significant reduction of the isoline contour area (ISOA) size between stimulus onset and offset (p=0.02, *Figure 3—figure supplement 1*, *Figure 4b* and *Video 3*). The median ISOA for stimulus onset locations was 92.5 arcmin$^2$ which was reduced to 68.2 arcmin$^2$ for stimulus offset locations. This decrease in size of the area of all retinal landing points supports the view of a certain retinal cone or a very small area of a few arcmin$^2$ to be the target region of the drift eye motion in a resolution task.

When we looked at how much the individual drift trajectory decreased the distance from either location, the median distance convergence (onset/offset distance) towards CDC, PRL, and PCD was about 12%, 7%, and 3%, respectively. While no participant had an average convergence of more than 30% towards PRL or PCD, the maximum convergence ratio towards CDC was about 50%. An adaptive drift behavior was also found in the relative drift lengths exhibited in each stimulus presentation. Although the individual drift lengths could vary substantially from trial to trial, we found that, across all participants and experimental trials, eyes exhibited significantly larger drift lengths when the stimulus onset location was further away from the CDC ($\rho^2$=0.06, p<0.001, *Figure 4g*). The onset distance was not correlated with drift directionality (*Figure 4g*). Across all trials, the average sampling cone density increased between stimulus onset and offset for most of the participants. This sampling gain was computed as the ratio between the maximum sampling density during the trial and the sampling density at the stimulus onset location. The sampling gain was significantly correlated with the potential retinal sampling gain of individuals in dominant (r$^2$=0.35, p=0.02) as well as non-dominant eyes (r$^2$=0.50, p=0.002, *Figure 4h*). Observers exploited on average 30% of their potential sampling gain in both fellow eyes. Interestingly, one observer combined all the previously described sampling features particularly strong in his dominant eye (P08_R). It had a steep cone density gradient, exhibited strong directional tuning towards the CDC, and had large drift lengths for stimulus onsets far from the CDC. This eye was excluded from the sampling gain analysis because fixation behavior differed by more than 4 standard deviations from the group average.

## Discussion

By using synchronous adaptive optics imaging and visual stimulation of the foveola, we find that the human visual system is capable of resolving spatial orientation of E optotypes smaller than a single photoreceptor diameter and uncover a fixational eye motor behavior that optimizes retinal sampling in accordance with the individual photoreceptor mosaic.

Spatial vision, and in particular visual acuity, is the most tested and used performance metric with a close relation to everyday vision. It provides the main behavioral outcome for clinical studies of vision. Measured in daily routine or clinical studies, the best corrected visual acuity of young and healthy adults is usually between 20/20 and 20/12.5 (60 and 37.5 arcsec) (*Reiniger et al., 2019*; *Rossi et al., 2007*). Even if lower order aberrations are corrected by e.g., glasses or contact lenses, higher order aberrations inherently blur the retinal image, depending on their magnitude (*Reiniger et al., 2019*). Adaptive optics induce a close-to-diffraction limited optical correction, where the optical improvement is significantly correlated with an increase in visual acuity thresholds (*Marcos et al., 2008*). By correcting aberrations with AOSLO, we measured Snellen-E thresholds that were up to half the size (between 20/10 and 20/6.9; 30–20.6 arcsec) compared to the natural viewing condition. This is slightly lower than previously presented data (*Rossi et al., 2007*), very likely because of the different wavelengths used for experimentation (*Figure 2c*). It might be surprising to learn that the neural machinery of human vision is able to resolve such tiny stimuli, because natural viewing is blurred by the eye's optics. Even though observers are, to some degree, adapted to their own aberrations (*Artal et al., 2004*), the best subjective image quality is seen when on average 88% of the aberrations are corrected (*Chen et al., 2007*). This may indicate that, under normal viewing conditions, optical aberrations and not cone topography may play the dominant role in limiting the eye's acuity.

By removing most aberrations in our experiments, we can study in how far resolution thresholds are linked to or limited by the optimized but at the same time individual morphology of the human foveola. While in the periphery, midget retinal ganglion cell sampling dominates resolution, resolution of the foveal center was estimated to be governed by the cone sampling limit (*Rossi and Roorda, 2010*; *Williams, 1985*). By first-time direct experimental validation in the same participants, we here confirm the hypothesis that the individual spacing of cones can predict the resolution capacity of our foveola when optical influences are bypassed (*Figure 2*). We found that the individual spatial arrangement of cones was highly correlated to the visual acuity of participants and explains up to 45% of its variance (*Figure 2d*). Eyes with higher foveolar sampling capacity reached lower thresholds than eyes with less densely packed cone photoreceptors. Moreover, all participants reached resolution thresholds that exceeded the static Nyquist sampling limit when tested with near-infrared, 788 nm light. Natural vision is comprised of multiwavelength stimuli, thus, using 788 nm in isolation is at the top end of our retinal sensitivity. In the first part of our study, participants also performed experiments with 840 nm light. Thresholds were rather approximating the Nyquist limit with this longer near-infrared wavelength (*Figure 2c*). The L- and M-cone photopigment absorbance for 840 nm is about 1.4 log unit lower than for 788 nm (*Stockman and Rider, 2023*). The decreased cone sensitivity combined with a larger Airy-Disk size of about 7% are likely to be detrimental for the longer, 840 nm, wavelength. We would expect a potential for even lower thresholds for shorter wavelengths.

Otherwise, a potential for lower thresholds is only expected in eyes with higher angular cone densities. Perhaps contrary at first sight, this could potentially be the case for observers with higher myopia. Myopic eyes, despite retinal stretching, generally have a higher angular sampling density in and around the foveola, compared to emmetropes (*Wang et al., 2019*). Therefore, we would expect acuity thresholds to be lower for myopic participants, in the case that (a) angular cone density is increased like previously suggested and (b) AO correction and display resolution are still sufficient to completely resolve the foveolar cone mosaic. Psychophysical data for more participants with higher myopia and longer axial lengths would be needed to verify this assumption.

Theoretical predictions of the Nyquist resolution limit imply stationary sampling. If the retinal image is under-sampled, aliasing occurs at the frequency of the receptor mosaic, which may obscure the original image, especially its orientation (compare example snapshots in *Figure 1f*). While prior knowledge of the stimulus has been shown to theoretically help to de-alias under-sampled signals even in a static condition (*Ruderman and Bialek, 1992*), we believe that retinal image motion plays a significant role in deciphering orientation at the limits of spatial sampling. Fixational eye movements continuously modulate the luminance flow on individual cones and postreceptoral neuronal activity. Drift motion has long been presumed as a random jitter, a result of the limited precision of the oculomotor system (*Cornsweet, 1956*; *Ditchburn and Ginsborg, 1953*). More recent work revealed that drift motion is neither random (*Rucci and Poletti, 2015*) nor detrimental due to the introduction of noise (*Burak et al., 2010*; *Pitkow et al., 2007*), but rather a fine-tuned motion, beneficial for psychophysical measures of visual acuity in the parafovea (*Ratnam et al., 2017*) as well as foveola (*Intoy and Rucci, 2020*). As also observed in other sensory organs (*Ahissar and Arieli, 2001*), neurons in the visual system are strongly selective not just for spatial patterns, but also for temporally changing stimuli (*Ahissar and Arieli, 2012*), a finding that is also supported by computational modeling, suggesting that the visual system may utilize principles comparable to those used in computational imaging for achieving super-resolution via camera motion (*Anderson et al., 2020*). Within the past decades, the interdisciplinary term 'geometrical super-resolution' which is devoted to the filtering properties of sensor systems has become common (*Zalevsky, 2011*). These resolution advantages may be achieved in the visual system by incorporating mechanisms that allow for the recognition of positional differences smaller than a single cell. That such mechanism exists is exemplified in a phenomenon known as hyperacuity. Fine localization discriminations of only a few seconds of arc are performed by identification of the centroid of the retinal light distributions (*Westheimer and McKee, 1977*) of the involved pattern components. In a diffraction-limited resolution task, the visual system seems to be able to translate the temporal luminance modulation in individual photoreceptors by ocular drift to additional spatial information about the stimulus position and shape. Contrary, the indirect suppression of natural fixational eye motion by retinal stabilization techniques impairs visual acuity outside the foveolar center (*Intoy and Rucci, 2020*; *Rucci et al., 2007*). For prolonged static stimulus presentations, retinal spiking decays over time, while drift motion keeps

the luminance change active, continuously refreshes the receptive field input, and sustains neuronal activity (*Kuang et al., 2012*).

We found a significant correlation between drift motion and visual acuity thresholds between individuals, indicating that drift motion may be one of the key elements in reaching sub-cone resolution thresholds. Interestingly, acuity improved for smaller fixational drift and decreased in participants who exhibited larger drift motion, on average. The fact that less drift is beneficial to reach the lowest possible acuity thresholds reflects the characteristics of spatiotemporal luminance changes introduced by smaller or larger drift motion. Smaller drifts induce luminance changes with higher spatial frequencies and models of retinal ganglion cell activity suggest a higher contrast sensitivity for high spatial frequency motion and less for low spatial frequencies compared to a static retina (*Kuang et al., 2012*; *Rucci and Victor, 2015*). This is supported by other recent work which also showed that visual acuity thresholds can even be predicted from drift magnitudes measured in a sustained fixation task (*Clark et al., 2022*).

There is evidence that fixational eye motion might have systematic components in primates. A previous study in macaque monkeys revealed a systematic directional drift response only a few dozens of milliseconds after various visual transients (*Malevich et al., 2020*). In our study, we reveal that a certain drift directionality can not only be triggered by particular visual transients, but that human observers are capable of adapting their drift direction to enact an oculomotor strategy that takes advantage of the maximum resolution capacity provided within the retina. Our participants precisely moved their eyes to have the stimulus slip across the most densely packed cone cells within their foveola. We hereby shed light on a mechanism that is potentially particularly active during fine discrimination tasks. This confirms that drift can be quickly adjusted in a continuous closed-loop control (*Gruber and Ahissar, 2020*), while, as other recent work suggests, being at the same time able to quickly switch to an open-loop process, as specific task knowledge influences the dominant orientation of drift, even in the sudden absence of visual information (*Lin et al., 2023*). Yet, the underlying neuronal control of drift motion remains not fully understood. Recent work suggested, based on brainstem recordings in rhesus monkeys, that the origin can be found mostly upstream of the ocular motoneurons. It can likely be explained as diffusion in the oculomotor integrator which is mainly driven by noise, but additionally affected by mechanisms within the visual motor pathway (e.g. feedback mechanisms) (*Ben-Shushan et al., 2022*). An incorporation of a visual feedback loop to that model was shown to modulate the statistics of eye motion, given a time lag of about 100 ms (mainly due to synaptic processing delays, of order 60–80ms *Malevich et al., 2020*). This fits our results well. Our presentation time of 500 ms sufficed for a modulation of the fixational drift motion towards retinal areas of higher cone sampling (also see *Figure 4—figure supplements 1–3*). Our data supports the view that some aspects of the statistics of drift motion can be influenced by the visual task (*Ben-Shushan et al., 2022*; *Intoy and Rucci, 2020*; *Malevich et al., 2020*; *Zhao et al., 2023*). The superior colliculus seems to play a major role in modulating drift motion in a feedback loop to visual inputs (*Hafed et al., 2021*). It's not only involved in controlling large eye motions (*Bergeron et al., 2003*) and microsaccades (*Hafed et al., 2009*), but also reflects neural responses to fixational drift that are likely a result of sensory input (C.-Y. *Chen et al., 2019*).

So, even though the CDC is displaced from the PRL in a way to be beneficial for natural binocular vision (*Reiniger et al., 2021*), constant visual feedback allows to adapt the drift direction and, therefore, also the task-related PRL. Commonly, the term PRL is used for describing the retinal location that is preferably used in fixational tasks. It is still a matter of debate what factors drive the development of this very reproducible (*Reiniger et al., 2021*) retinal location and in how far it might provide enhanced visual function. Sensitivity to small light spots in the foveola seems to be rather plateau-like and not particularly pronounced at the PRL (*Domdei et al., 2021*). As recently shown, the PRL slightly differs between different tasks but has a larger interindividual variability (*Bowers et al., 2021*). The here shown results indicate that also when measuring visual resolution, the PRL is not necessarily the center of the sampling drift motion. The directional drift motion leads to a shift of the preferred retinal location for a resolution task towards the CDC (*Figure 4—figure supplement 3* and *Video 3*). Previous work that compared active versus passive fixation did not show a systematic offset in a similar experimental setup. However, 5 out of 8 participants also shifted their PRL in a Snellen E task closer to the CDC compared to the PRL for fixating a static disk stimulus (*Bowers et al., 2021*), the conditions that are best comparable to our study. The main difference to

our visual acuity experiments was that automatically paced random time intervals between presentations (0.5–1.5 s) were applied to not allow the participants to anticipate the next trial whereas in our study, participants self-paced the stimulus output to be able to prepare and focus for the next trial. It might be that this extremely fine-tuned usage of the visual feedback loop can only be kept active for rather short time intervals. By shifting the stimulus towards the CDC in 50% of cases, the potential sampling gain within individual eyes was exploited by 30%, on average, which goes along with a cone density increase of 3% or 285 cones/deg$^2$. Even though this increase in cone density alone would not account for the difference between acuity thresholds and the Nyquist limit, this and the simultaneous spatiotemporal luminance modulation contribute to achieving sub-cone visual acuity thresholds.

Between fellow eyes, we found very strong correlations for all the measured parameters. While drift lengths and directionality, as well as cone densities are very symmetric between dominant and non-dominant eyes (*Figures 2e and 3e*), significantly lower acuity thresholds of 1.5 arcsec, on average, were observed in the dominant eyes of participants (*Figure 2e*). The dominant eyes' visual input has a tendency to be preferred during binocular viewing, but has not been shown to exhibit relevant differences in visual function in healthy eyes with low refractive errors (*Ehrenstein et al., 2005*; *Zhou et al., 2017*). Partially this may be due to limited accuracy in the mainly used clinical methods (e.g. Snellen Chart or projection have ~10 arcsec steps between optotype rows). This very fine binocular difference between eyes emphasizes that some remaining factors which especially comprise the neural postprocessing steps, also play an important role and may facilitate the slight functional advantage of dominant eyes.

For clinical studies of retinal health and in new therapeutical approaches, photoreceptor health and visual acuity can be related to other more standard clinical measures as OCT-derived measures of outer segment length or retinal thickness which have been shown to serve for estimates of cone density (*Domdei et al., 2023*). Therefore, building a larger dataset on photoreceptor-resolved foveolar maps and associated visual function measures may help to, on the one hand, better understand the interplay between structural and functional changes to draw conclusions about disease progression, intervention efficiency, or the interpretation of retinal imaging data in studies aimed at vision restoration. On the other hand, a detailed examination of psychophysical measures with knowledge about the exact neural sampling characteristics offers a great potential to answer further questions about e.g., resolution limits in myopia, the effect of image stabilization in the very center of the foveola, or implications for binocular viewing that could previously only be hypothesized. The awareness of the oculomotor system being able to finely adjust the drift motion behavior for a particular task may guide future interpretation of fixational eye motion.

# Materials and methods

## Participants

A total of 38 participants with White ethnicity underwent a preliminary screening where ocular biometry, ophthalmologic status, fixational eye motion, and adaptive optics correction as well as foveolar image quality were tested. From those, 20 participants with normal ophthalmologic status, resolvable foveolar cones, and ocular anatomy that allowed for a 7 mm pupil aperture during experimentation were chosen for subsequent examination. All 6 male and 14 female observers (17 adults [age: 18–42], three children [age: 10, 12, and 14]) had no or only mild refractive errors (SE:±2.5 diopters). The children and 15 adults were naïve participants and two adults were experienced observers. More detailed cone topography and fixational eye motion characteristics of the here studied population have been shown previously (*Reiniger et al., 2021*). The experiments were conducted under two different light conditions (16 participants 788 nm, 12 participants 840 nm). Eight participants took part in both experimental conditions. We mainly report the data acquired for the 788 nm condition in this manuscript and show 840 nm data for comparison where noteworthy differences arise.

Written informed consent was obtained from each participant and all experimental procedures adhered to the tenets of the Declaration of Helsinki, in accordance with the guidelines of the independent ethics committee of the medical faculty at the Rheinische Friedrich-Wilhelms-Universität of Bonn.

## Ocular dominance

Ocular dominance was determined by a Miles Test prior to pupil dilation and visual acuity testing. The experimenter stood at a distance of 6 m in front of the participants and asked them to form a small opening between their thumbs and forefingers with both hands. The participant was then asked to extend their arms in front of them to look through the formed hole in the experimenter's face with both eyes open. This procedure was conducted three to five times to determine the dominant (=uncovered) eye in a 3/3 or at least 3/5 condition.

## AOSLO retinal imaging

In vivo images of the complete foveolar cone mosaic were recorded using a custom-built adaptive optics scanning laser ophthalmoscope (AOSLO). The general setup of the AOSLO has been described previously (*Roorda et al., 2002*) and pertinent differences as well as the method of determination of the preferred retinal locus of fixation (PRL) have been described in a recent publication (*Reiniger et al., 2021*).

In brief, the front end of the AOSLO was equipped with three *f*=500 mm focal telescopes. These telescopes were specifically designed for point-scanning an adaptive optics-corrected focal light spot across the retina, ensuring diffraction-limited resolution in both incident and reflected beams. The system incorporated a magnetic actuator-driven deformable mirror (DM97-07, 7.2 mm pupil diameter, ALPAO, Montbonnot-Saint-Martin, France) positioned in a retinal conjugate plane. The deformable mirror was controlled by the wavefront error signals from a 25×25 lenslet Shack Hartmann sensor (SHSCam AR-S-150-GE, Optocraft GmbH, Erlangen, Germany) in closed-loop. Imaging and wave-front correction utilized wavelengths of either 788 nm (±12 nm) or 840 nm (±12 nm) light, achieved through serial dichroic and bandpass filtering of a supercontinuum source (SuperK Extreme EXR-15, NKT Photonics, Birkerød, Denmark). The imaging field of view was 0.85×0.85 degrees of visual angle. The digital lateral resolution was about 0.1 arcmin, the size of one pixel in the recorded videos and images. Light reflected from the retina was detected by a photomultiplier tube (PMT, H7422-50, Hamamatsu Photonics, Hamamatsu, Japan), positioned behind a confocal pinhole (Pinhole diameter = 20 mm, equivalent to 0.47 (840 nm) and 0.5 (788 nm) Airy disk diameters). Continuous sampling of the PMT signal was carried out using a field programmable gate array (FPGA), resulting in a 512x512-pixel video at 30 Hz (600 pixels per degree of visual angle). Through rapid acousto-optic intensity modulation of the imaging lights, the square AOSLO imaging field was used as a retinal display, where each pixel could be individually controlled to produce the visual stimuli.

## Cone map generation and computation of sampling characteristics

The best PRL videos acquired were selected to create spatially registered, high signal-to-noise ratio images of the foveal center, which served as master retinal images for cone labeling as well as referencing of stimulus motion trajectories. This study includes only participants for whom the master retinal image was of sufficient quality to label all cones across the image. Cone centers were identified and labeled semi-manually, as previously described (*Cunefare et al., 2017*; *Reiniger et al., 2021*). Cone density was computed in two different ways. First, for deriving landmark metrics of the foveolar cone map, we then computed Voronoi tessellation, estimating a patch with a certain area for each individual cone and summed the nearest 150 cone patches around each image pixel. The number of cells was divided by the resulting area to derive a pixel-resolved map of cone densities. Based on this map, the peak cone density (PCD) is defined as the highest cone density value of the map with its according retinal location. The cone density centroid (CDC) is computed as the weighted centroid of the 20th percentile of the highest cone densities within the map. We refer to the CDC as the anatomical center and the anchor for further spatial analyses in this study. The CDC has been shown to be a more robust and reproducible metric to describe the anatomical center than the more routinely reported PCD. While the PCD has value in reporting its quantity, namely the maximum cone density of a retina, using it as a landmark is however not advised, for it is too vulnerable against small changes in the analysis of cone density (*Reiniger et al., 2021*; *Wynne et al., 2022*).

Second, for analyzing the relation between individual sampling limits and resolution acuity, cone density was computed based on the cone cells contributing to the sampling process. To identify the cones interacting in stimulus sampling, a simple model of cone light capture was employed. Each cone was described by an associated light acceptance aperture with its diameter estimated as 48% of

the average spacing between the cone and all of its neighbors. The efficiency of the aperture along its diameter was approximated as Gaussian profiles. Also, a model of the stimulus retinal image was computed by convolving the eye's point spread function (diffraction limited at 788 nm for a 7 mm pupil) with the stimulus bitmap. The complete two-dimensional model of cone apertures was then multiplied by models of the presented stimuli to arrive at the cone-level light distribution based on the different stimulus positions, sizes, and orientations. The light distribution within each cone was integrated across the entire cone aperture. This value was then normalized to the degree to which the aperture was filled. Cone stimulation was considered to be maximal if the entire aperture was filled. Using this method, a cone activation pattern could be generated for each point in time (e.g. *Figure 1f*). To arrive at a task-related cone density estimate for each frame (sampling cone density), the number of cones identified to interact with the stimulus was divided by their summed cone area. In the presented analyses, the median sampling density of all trials is analyzed and standard deviations are shown as gray lines (*Figure 2d and e*). This stimulus-related cone density was chosen to closely represent the sampling process; however, the results do not qualitatively differ from using the cone density map based on the 150 nearest cones.

We assumed a perfect hexagonal cell mosaic to estimate the average inter-cone-distance (ICD) between neighboring cells and to compute the theoretical Nyquist sampling limit, which is based on the spacing between rows of cones, and given by $N = \frac{\sqrt{3}}{2} \times \text{ICD}$.

## Experimental procedures

For psychophysical acuity testing, participants reported the orientation of a Snellen-E stimulus in a four-alternative forced-choice (four AFC) task under unrestricted eye motion. Psychophysical experiments were performed monocularly in both eyes. The non-dominant eye was tested first and the dominant eye after a 15–30 min break. This protocol was chosen because with pilot experiments in seven participants (which were performed in a random order) less time was needed and hence less fatigue was reported by the participants when the second eye was the dominant one. In these pilot experiments, the same qualitative difference of acuity thresholds between non-dominant and dominant were found.

Mydriasis and cycloplegia were established by two drops of 1% tropicamide, instilled into the eyelid about 25 and 20 min prior to experiments. If experimentation took longer than 40 min, another drop of tropicamide was instilled. A customized dental impression mold (bite bar) was used to immobilize and adjust the head position and thus to align the participant's eye in front of the imaging system to ensure optimal adaptive optics correction and image quality. The participants were encouraged to take breaks at any time. We found that proper resting is one of the most crucial factors during the rather complex AOSLO experimentation. Frequent breaks ensure constant, high-level compliance and excellent image quality as the basis for artefact-free and reproducible results.

Before recording experimental runs, each participant performed three test runs to get used to the test procedure and the appearance of the stimuli. The stimuli were displayed as 'off-stimuli' on the infrared background by switching the displayed intensity via an acousto-optic modulator (*Poonja et al., 2005*) (AOM, TEM-250-50-10-840-2FP, Brimrose, Sparks Glencoe, MD, USA) (*Figure 1a*). Because of ocular diffraction, the stimulus contrast varied between 0.61 and 0.80 for an 18 arcsec versus 36 arcsec gap-sized stimulus (three and six pixels of the scanning raster, respectively). The visual acuity testing followed the Bayesian adaptive procedure QUEST (*Brainard, 1997*; *Pelli, 1997*; *Watson and Pelli, 1983*). Stimulus progression was self-paced by the participant. The stimuli were presented for 500 ms to avoid limitations by insufficient temporal summation (*McAnany, 2014*). Around each trial, a 1 s AOSLO video was recorded, with the stimulation onset at around 300 ms after video onset. Visual acuity thresholds were estimated by pooling results from 5 consecutively run staircases, with each containing 20 trials. A psychometric function was fitted using *psignifit4* (*Schütt et al., 2016*) to derive threshold estimates for further analysis. The expected threshold variance is described and visualized by the 95% confidence interval (*Figures 1d, e , and 2d*).

## Video processing and eye motion analysis

The AOSLO used a raster scanning technique where each frame was acquired over time. The recorded videos were stabilized after psychophysical testing using custom settings within the *MATLAB*-based stabilization software from Stevenson et al. (*Stevenson and Roorda, 2005*). To acquire eye traces at

higher temporal resolution than the 30 Hz frame rate, each frame of the AOSLO movie is broken into 32 horizontal strips of 16 pixels height and cross-correlated against a reference frame. The reference frame was generally chosen automatically and exchanged by a manually chosen frame in cases weres stabilization failed despite good overall image quality. This method allowed the extraction of eye motion traces at temporal frequencies up to 960 Hz.

The frame-wise (30 Hz) stimulus position was encoded as a white cross marker in each video. As single-strip alignments can have small errors due to noise in the strip or retinal torsion (particularly affecting the horizontal motion estimate) (*Hofmann et al., 2022*), we compute the average offsets from the cross-containing strip and two previous/subsequent strips. These steps yielded more accurate trajectories in retinal coordinates for every trial. All individual trial AOSLO frames and the corresponding trajectories are then referenced to the single master retinal image used for cone map generation.

To quantify the retinal motion across the stimulus, drift length was defined as the concatenated vector sum of all frame-wise motion vectors within the 500 ms stimulus duration (see also *Figure 3c*). Trials that contained microsaccades or blinks during stimulus presentation were excluded from further analyses. Microsaccade occurrence varied highly between participants (mean ± SD: 14±10% of trials, range: 2-41%). If not stated differently, we here report the median drift length of all trials for individual eyes (e.g. of all traces shown in *Figure 3a*). To quantify drift direction, the angle between each trajectory's starting coordinate (coordinate center in *Figure 4c*) and end coordinate was computed. To check for potential motion bias, the drift angles were first analyzed in retinal coordinates (*Figure 4c*), and then as the relative angle, $\theta_{CDC}$, formed between the drift vector and the line connecting the retinal onset location and the CDC (*Figure 4d*). To compare directionality towards other locations of interest, the same was done for PRL and PCD locations (*Figure 4f*).

## Statistical information

All statistical analyses were conducted using custom-written MATLAB code and significance levels were set at 0.05. To assess the normal distribution of the dataset, a two-sided Shapiro–Wilk test was employed. This test is recognized to be appropriate for small sample sizes. The paired samples t-test was utilized to assess whether there were significant differences between the means of normally distributed paired observations. For non-parametric data, the Wilcoxon Signed-Rank test was employed. Linear correlations were computed to examine the relationships between variables. For variables demonstrating normal distribution, Pearson's correlation coefficient was employed, while for non-normally distributed data, Spearman's rank correlation coefficient was utilized. Pearson's correlation is sensitive to linear relationships, assuming bivariate normality, whereas Spearman's correlation is a non-parametric measure suitable for monotonic relationships and is robust against outliers and non-normal distributions.

## Acknowledgements

This work is supported by the German Research Foundation, Emmy Noether-Program HA5323/5-1; Carl Zeiss Foundation, HC-AOSLO; Novartis Pharma GmbH, EYENovative research award, and the Open Access Publication Fund of the University of Bonn.

## Additional information

### Funding

| Funder | Grant reference number | Author |
| --- | --- | --- |
| Deutsche Forschungsgemeinschaft | HA5323/5-1 | Wolf M Harmening |
| Carl Zeiss Foundation | HC-AOSLO | Wolf M Harmening |
| Novartis Pharma GmbH | EYENovative research award | Jenny L Witten |

| Funder | Grant reference number | Author |
|---|---|---|
| University of Bonn | Open Access Publication Fund | Jenny L Witten |

The funders had no role in study design, data collection and interpretation, or the decision to submit the work for publication.

## Author contributions

Jenny L Witten, Conceptualization, Resources, Data curation, Software, Formal analysis, Investigation, Visualization, Methodology, Writing – original draft, Project administration, Writing – review and editing; Veronika Lukyanova, Resources, Software, Formal analysis; Wolf M Harmening, Conceptualization, Resources, Supervision, Funding acquisition, Validation, Writing – original draft, Project administration, Writing – review and editing

## Author ORCIDs

Jenny L Witten ⓘ https://orcid.org/0000-0001-7500-7053
Veronika Lukyanova ⓘ https://orcid.org/0009-0008-3682-7225
Wolf M Harmening ⓘ https://orcid.org/0000-0001-7053-1198

## Ethics

Participants gave written informed consent before the experiments. All studies complied with the Declaration of Helsinki in its latest version and were approved by the Ethics Committee of the medical faculty at the Rheinische Friedrich-Wilhelms-Universität of Bonn (reference number: 2018-09).

Reviewer #1 (Public review): https://doi.org/10.7554/eLife.98648.3.sa1
Reviewer #2 (Public review): https://doi.org/10.7554/eLife.98648.3.sa2
Reviewer #3 (Public review): https://doi.org/10.7554/eLife.98648.3.sa3
Author response https://doi.org/10.7554/eLife.98648.3.sa4

# Additional files

## Supplementary files

• MDAR checklist

## Data availability

The following data are publicly available for download at Mendeley Data: 1. All original cone mosaic images and cone coordinates. 2. Retinal locations of CDC, PRL and PCD. 3. Fixational eye motion trajectories. 4. MATLAB code that can be used to plot the data on the original image.

The following dataset was generated:

| Author(s) | Year | Dataset title | Dataset URL | Database and Identifier |
|---|---|---|---|---|
| Witten JL, Lukyanova V, Harmening W | 2024 | Data from: Sub-cone visual resolution by active, adaptive sampling in the human foveola | https://data.mendeley.com/datasets/zp6d5w8kdv/1 | Mendeley Data, 10.17632/zp6d5w8kdv.1 |

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
