## [Editor Report · eLife Assessment]

This **important** work uses in vivo foveal cone-resolved imaging and simultaneous microscopic photostimulation to investigate the relationship between ocular drift - eye movements long thought to be random - and visual acuity. The surprising result is that ocular drift is systematic - causing the object to move to the center of the cone mosaic over the course of each perceptual trial. The tools used to reach this conclusion are state-of-the-art and the evidence presented is **convincing**. This work advances our understanding of the visuomotor system and the interplay of anatomy, oculomotor behavior, and visual acuity.

---

## [Referee Report · Reviewer #1 (Public review)]

Summary:

This paper investigates the relationship between ocular drift - eye movements long thought to be random - and visual acuity. This is a fundamental issue for how vision works. The work uses adaptive optics retinal imaging to monitor eye movements and where a target object is in the cone photoreceptor array. The surprising result is that ocular drift is systematic - causing the object to move to the center of the cone mosaic over the course of each perceptual trial. The tools used to reach this conclusion are state-of-the-art and the evidence presented is convincing.

Strengths

The central question of the paper is interesting, as far as I know, it has not been answered in past work, and the approaches employed in this work are appropriate and provide clear answers.

The central finding - that ocular drift is not a completely random process - is important and has a broad impact on how we think about the relationship between eye movements and visual perception.

The presentation is quite nice: the figures clearly illustrate key points and have a nice mix of primary and analyzed data, and the writing (with one important exception) is generally clear.

Weaknesses

The primary concern I had about the previous version of the manuscript was how the Nyquist limit was described. The changes the authors made have improved this substantially in the current version.

---

## [Referee Report · Reviewer #2 (Public review)]

Summary:

In this work, Witten et al. assess visual acuity, cone density, and fixational behavior in the central foveal region in a large number of subjects.

This work elegantly presents a number of important findings, and I can see this becoming a landmark work in the field. First, it shows that acuity is determined by the cone mosaic, hence, subjects characterized by higher cone densities show higher acuity in diffraction limited settings. Second, it shows that humans can achieve higher visual resolution than what is dictated by cone sampling, suggesting that this is likely the result of fixational drift, which constantly moves the stimuli over the cone mosaic. Third, the study reports a correlation between the amplitude of fixational motion and acuity, namely, subjects with smaller drifts have higher acuities and higher cone density. Fourth, it is shown that humans tend to move the fixated object toward the region of higher cone density in the retina, lending further support to the idea that drift is not a random process, but is likely controlled. This is a beautiful and unique work that furthers our understanding of the visuomotor system and the interplay of anatomy, oculomotor behavior, and visual acuity.

Strengths:

The work is rigorously conducted, it uses state-of-the-art technology to record fixational eye movements while imaging the central fovea at high resolution, and examines exactly where the viewed stimulus falls on individuals' foveal cone mosaic with respect to different anatomical landmarks in this region. Figures are clear and nicely packaged. It is important to emphasize that this study is a real tour-de-force in which the authors collected a massive amount of data on 20 subjects. This is particularly remarkable considering how challenging it is to run psychophysics experiments using this sophisticated technology. Most of the studies using psychophysics with AO are, indeed, limited to a few subjects. Therefore, this work shows a unique set of data, filling a gap in the literature.

Weaknesses:

Data analysis has been improved after the first round of review. The revised version of the manuscript is solid, and there are no weaknesses that should be addressed. The authors added more statistical tests and analyses, reported comparable effects even when different metrics are used (e.g., diffusion constant), and removed the confusing text on myopia. I think this work represents a significant scientific contribution to vision science.

---

## [Referee Report · Reviewer #3 (Public review)]

Summary:

The manuscript by Witten et al., aims to investigate the link between acuity thresholds (and hyperacuity) and retinal sampling. Specifically, using in vivo foveal cone-resolved imaging and simultaneous microscopic photo stimulation, the researchers examined visual acuity thresholds in 16 volunteers and correlated them with each individual's retinal sampling capacity and the characteristics of ocular drift.

First, the authors found that although visual acuity was highly correlated with the individual spatial arrangement of cones, for all participants, visual resolution exceeded the Nyquist sampling.

Thus, the researchers hypothesized that this increase in acuity, which could not be explained in terms of spatial encoding mechanisms, might result from exploiting the spatiotemporal characteristics of the visual input associated with the dynamics of the fixational eye movements (and ocular drift in particular).

The authors reported a correlation between acuity threshold and drift amplitude, suggesting that the visual system benefits from transforming spatial input into a spatiotemporal flow. Finally, they showed that drift, contrary to the traditional view of it as random involuntary movement, appears to exhibit directionality: drift tends to move stimuli to higher cone density areas, therefore enhancing visual resolution.

I find the work of broad interest, its methods are clear, and the results solid.

---

## [Author Response]

The following is the authors’ response to the original reviews.

**Reviewer #1 (Public Review):**
Summary:This paper investigates the relationship between ocular drift - eye movements long thought to be random - and visual acuity. This is a fundamental issue for how vision works. The work uses adaptive optics retinal imaging to monitor eye movements and where a target object is in the cone photoreceptor array. The surprising result is that ocular drift is systematic - causing the object to move to the center of the cone mosaic over the course of each perceptual trial. The tools used to reach this conclusion are state-of-the-art and the evidence presented is convincing.StrengthsP1.1. The central question of the paper is interesting, as far as I know, it has not been answered in past work, and the approaches employed in this work are appropriate and provide clear answers.P1.2. The central finding - that ocular drift is not a completely random process - is important and has a broad impact on how we think about the relationship between eye movements and visual perception.P1.3. The presentation is quite nice: the figures clearly illustrate key points and have a nice mix of primary and analyzed data, and the writing (with one important exception) is generally clear.

Thank you for your positive feedback.

WeaknessesP1.4. The handling of the Nyquist limit is confusing throughout the paper and could be improved. It is not clear (at least to me) how the Nyquist limit applies to the specific task considered. I think of the Nyquist limit as saying that spatial frequencies above a certain cutoff set by the cone spacing are being aliased and cannot be disambiguated from the structure at a lower spatial frequency. In other words, there is a limit to the spatial frequency content that can be uniquely represented by discrete cone sampling locations. Acuity beyond that limit is certainly possible with a stationary image - e.g. a line will set up a distribution of responses in the cones that it covers, and without noise, an arbitrarily small displacement of the line would change the distribution of cone responses in a way that could be resolved. This is an important point because it relates to whether some kind of active sampling or movement of the detectors is needed to explain the spatial resolution results in the paper. This issue comes up in the introduction, results, and discussion. It arises in particular in the two Discussion paragraphs starting on line 343.

We thank you for pointing out a possible confusion for readers. Overall, we contrast our results to the static Nyquist limit because it is generally regarded as the upper limit of resolution acuity. We updated our text in a few places, especially the Discussion, and added a reference to make our use of the Nyquist limit clearer.

We agree with the reviewer of how the Nyquist limit is interpreted within the context of visual structure. If visual structure is under-sampled, it is not lost, but creates new, interfered visual structure at lower spatial frequency. For regular patterns like gratings, interference patterns may emerge akin to Moire patterns, which have been shown to occur in the human eye, and which form is based on the arrangement and regularity of the photoreceptor mosaic (Williams, 1985). We note however that the successful resolution of the lower frequency pattern does not necessarily carry the same structural information, specifically, orientation, and the aliased structure might indeed mask the original stimulus. Please compare Figure 1f where we show individual static snapshots of such aliased patterns, especially visible when the optotypes are small (towards the lower right of the figure). We note that theoretical work predicts that with prior knowledge about the stimulus, even such static images might be possible to de-alias (Ruderman & Bialek, 1992). We added this to our manuscript.

We think the reviewer’s following point about the resolution of a line position, is only partially connected to the first, however. In our manuscript we note in the Introduction that resolution of the relative position of visual objects is a so called hyperacuity phenomenon. The fact that it occurs in humans and other animals demonstrates that visual brains have come up with neuronal mechanisms to determine relative stimulus position with sub-Nyquist resolution. The exact mechanism is however not fully clear. One solution is that relative cone signal intensities could be harnessed, similar as is employed technically, e.g. in a quadrant-cell detector. Its positional precision is much higher than the individual cell’s size (or Nyquist limit), predominantly determined by the detector’s sensitivity and to a lesser degree its size. On the other hand, such detector, being hyperacute with object location, would not have the same resolution as, for instance, letter-E orientation discrimination.

Note that in all the above occasions, a static image-sensor-relationship is assumed. In our paper, we were aiming to convey, like others did before, that a moving stimulus may give rise to sub-Nyquist structural resolution, beyond what is already known for positional acuity and hence, classical hyperacuity.

Based on the data shown in this manuscript and other experimental data currently collected in the lab, it seems to us that eye movements are indeed the crucial point in achieving sub-Nyquist resolution. For example, ultra-short presentation durations, allowing virtually no retinal slip, push thresholds close to the Nyquist limit and above. Furthermore, with AOSLO stimulation, it is possible to stabilize a stimulus on the retina, which would be a useful tool studying this hypothesis. Our current level of stabilization is however not accurate enough to completely mitigate retinal image motion in the foveola, where cells are smallest, and transients could occur. From what we observe and other studies that looked at resolution thresholds at more peripheral retinal locations, we would predict that foveolar resolution of a perfectly stabilized stimulus would be indeed limited by the Nyquist limit of the receptor mosaic.

P1.5. One question that came up as I read the paper was whether the eye movement parameters depend on the size of the E. In other words, to what extent is ocular drift tuned to specific behavioral tasks?

This is an interesting question. Yet, the experimental data collected for the current manuscript does not contain enough dispersion in target size to give a definitive answer, unfortunately. A larger range of stimulus sizes and especially a similar number of trials per size would be required. Nonetheless, when individual trials were re-grouped to percentiles of all stimulus sizes (scaled for each eye individually), we found that drift length and directionality was not significantly different between any percentile group of stimulus sizes (Wilcoxon sign rank test, p > 0.12, see also Figure R1). Our experimental trials started with a stimulus demanding visual acuity of 20/16 (logMAR = -0.1), therefore all presented stimulus sizes were rather close to threshold. The high visual demand in this AO resolution task might bring the oculomotor system to a limit, where ocular drift length can’t be decreased further. However, with the limitation due to the small range of stimulus sizes, further investigations would be needed. Given this and that this topic is also ongoing research in our lab where also more complex dynamics of FEM patterns are considered, we refrain from showing this analysis in the current manuscript.

**Author response image 1. sa4fig1:** Drift length does not depend on stimulus sizes close to threshold. All experimental trials were sorted by stimulus size and then grouped into percentiles for each participant (left). Additionally, 10 % of trials with stimulus sizes just above or below threshold are shown for comparison (right). For each group, median drift lengths (z-scored) are shown as box and whiskers plot. Drift length was not significantly different across groups.

**Reviewer #2 (Public Review):**
Summary:In this work, Witten et al. assess visual acuity, cone density, and fixational behavior in the central foveal region in a large number of subjects.This work elegantly presents a number of important findings, and I can see this becoming a landmark work in the field. First, it shows that acuity is determined by the cone mosaic, hence, subjects characterized by higher cone densities show higher acuity in diffraction-limited settings. Second, it shows that humans can achieve higher visual resolution than what is dictated by cone sampling, suggesting that this is likely the result of fixational drift, which constantly moves the stimuli over the cone mosaic. Third, the study reports a correlation between the amplitude of fixational motion and acuity, namely, subjects with smaller drifts have higher acuities and higher cone density. Fourth, it is shown that humans tend to move the fixated object toward the region of higher cone density in the retina, lending further support to the idea that drift is not a random process, but is likely controlled. This is a beautiful and unique work that furthers our understanding of the visuomotor system and the interplay of anatomy, oculomotor behavior, and visual acuity.Strengths:P2.1. The work is rigorously conducted, it uses state-of-the-art technology to record fixational eye movements while imaging the central fovea at high resolution and examines exactly where the viewed stimulus falls on individuals' foveal cone mosaic with respect to different anatomical landmarks in this region. The figures are clear and nicely packaged. It is important to emphasize that this study is a real tour-de-force in which the authors collected a massive amount of data on 20 subjects. This is particularly remarkable considering how challenging it is to run psychophysics experiments using this sophisticated technology. Most of the studies using psychophysics with AO are, indeed, limited to a few subjects. Therefore, this work shows a unique set of data, filling a gap in the literature.

Thank you, we are very grateful for your positive feedback.

Weaknesses:P2.2. No major weakness was noted, but data analysis could be further improved by examining drift instantaneous direction rather than start-point-end-point direction, and by adding a statistical quantification of the difference in direction tuning between the three anatomical landmarks considered.

Thank you for these two suggestions. We now show the development of directionality with time (after the first frame, 33 ms as well as 165 ms, 330 ms and 462 ms), and performed a Rayleigh test for non-uniformity of circular data. Please also see our response to comment R2.4.

Briefly, directional tuning was already visible at 33 ms after stimulus onset and continuously increases with longer analysis duration. Directionality is thus not pronounced at shorter analysis windows. These results have been added to the text and figures (Figure 4 - figure supplement 1).

The statistical tests showed that circular sample directionality was not uniformly distributed for all three retinal locations. The circular average was between -10 and 10 ° in all cases and the variance was decreasing with increasing time (from 48.5 ° to 34.3 ° for CDC, 49.6 ° to 38.6 ° for PRL and 53.9 ° to 43.4 for PCD location, between frame 2 and 15). As we have discussed in the paper, we would expect all three locations to come out as significant, given their vicinity to the CDC (which is systematic in the case of PRL, and random in the case of PCD, see also comment R2.2).

**Reviewer #3 (Public Review):**
Summary:The manuscript by Witten et al., titled "Sub-cone visual resolution by active, adaptive sampling in the human foveola," aims to investigate the link between acuity thresholds (and hyperacuity) and retinal sampling. Specifically, using in vivo foveal cone-resolved imaging and simultaneous microscopic photostimulation, the researchers examined visual acuity thresholds in 16 volunteers and correlated them with each individual's retinal sampling capacity and the characteristics of ocular drift.First, the authors found that although visual acuity was highly correlated with the individual spatial arrangement of cones, for all participants, visual resolution exceeded the Nyquist sampling limit - a well-known phenomenon in the literature called hyperacuity.Thus, the researchers hypothesized that this increase in acuity, which could not be explained in terms of spatial encoding mechanisms, might result from exploiting the spatiotemporal characteristics of visual input, which is continuously modulated over time by eye movements even during so-called fixations (e.g., ocular drift).Authors reported a correlation between subjects, between acuity threshold and drift amplitude, suggesting that the visual system benefits from transforming spatial input into a spatiotemporal flow. Finally, they showed that drift, contrary to the traditional view of it as random involuntary movement, appears to exhibit directionality: drift tends to move stimuli to higher cone density areas, therefore enhancing visual resolution.Strengths:P3.1. The work is of broad interest, the methods are clear, and the results are solid.

Thank you.

Weaknesses:P3.2. Literature (1/2): The authors do not appear to be aware of an important paper published in 2023 by Lin et al. (https://doi.org/10.1016/j.cub.2023.03.026), which nicely demonstrates that (i) ocular drifts are under cognitive influence, and (ii) specific task knowledge influences the dominant orientation of these ocular drifts even in the absence of visual information. The results of this article are particularly relevant and should be discussed in light of the findings of the current experiment.

Thank you for pointing to this important work which we were aware of. It simply slipped through during writing. It is now discussed in lines 390-393.

P3.3. Literature (2/2): The hypothesis that hyperacuity is attributable to ocular movements has been proposed by other authors and should be cited and discussed e.g., https://doi.org/10.3389/fncom.2012.00089, https://doi.org/10.10

Thank you for pointing us towards these works which we have now added to the Discussion section. We would like to stress however, that we see a distinction between classical hyperacuity phenomena (Vernier, stereo, centering, etc.) as a form of positional acuity, and orientation discrimination.

P3.4. Drift Dynamic Characterization: The drift is primarily characterized as the "concatenated vector sum of all frame-wise motion vectors within the 500 ms stimulus duration.". To better compare with other studies investigating the link between drift dynamics and visual acuity (e.g., Clark et al., 2022), it would be interesting to analyze the drift-diffusion constant, which might be the parameter most capable of describing the dynamic characteristics of drift.

During our analysis, we have computed the diffusion coefficient (D) and it showed qualitatively similar results to the drift length (see figures below). We decided to not show these results, because we are convinced that D is indeed not the most capable parameter to describe the typical drift characteristic seen here. The diffusion coefficient is computed as the slope of the mean square displacement (MSD). In our view, there are two main issues with applying this metric to our data, one conceptual, one factual:

(1) Computation of a diffusion coefficient is based upon the assumption that the underlying movement is similar to a random walk process. From a historical perspective, where drift has been regarded as more random, this makes sense. We also agree that D can serve as a valuable metric, depending on the individual research question. In our data, however, we clearly show that drift is not random, and a metric quantifying randomness is thus ill-defined.

(2) We often observed out- and in-type motion traces, i.e. where the eye somewhat backtracks from where it started. Traces in this case are equally long (and fast) as other motion will be with a singular direction, but D would in this case be much smaller, as the MSD first increases and then decreases. In reality, the same number of cones would have been traversed as with the larger D of straight outward movement, albeit not unique cones. For our current analyses, the drift length captures this relationship better.

**Author response image 2. sa4fig2:** Diffusion coefficient (D) and the relation to visual acuity (see Figure 3 e-g for comparison to drift length). (a) D was strongly correlated between fellow eyes. (b) Cone density and D were not significantly correlated. (c) The median D had a moderate correlation with visual acuity thresholds in dominant as well as non-dominant eyes. Dominant eyes are indicated by filled, nondominant eyes by open markers.

We would like to put forward that, in general, better metrics are needed, especially in respect to the visual signals arising from the moving eye. We are actively looking into this in follow-up work, and we hope that the current manuscript might spark also others to come up with new ways of characterizing the fine movements of the eye during fixation.

P3.5. Possible inconsistencies: Binocular differences are not expected based on the hypothesis; the authors may speculate a bit more about this. Additionally, the fact that hyperacuity does not occur with longer infrared wavelengths but the drift dynamics do not vary between the two conditions is interesting and should be discussed more thoroughly.

Binocularity: the differences in performance between fellow eyes is rather subtle, and we do not have a firm grip on differences other than the cone mosaic and fixational motor behavior between the two eyes. We would rather not speculate beyond what we already do, namely that some factor related to the development of ocular dominance is at play. What we do show with our data is that cone density and drift patterns seem to have no part in it.

Effect of wavelength: even with the longer 840 nm wavelength, most eyes resolve below the Nyquist limit, with a general increase in thresholds (getting worse) compared to 788 nm. As we wrote in the manuscript, we assume that the increased image blur and reduced cone contrast introduced by the longer wavelength are key to why there is an overall reduction in acuity. No changes were made to the manuscript. As a more general remark, we would not consider the sub-Nyquist performances seen in our data to be a hyperacuity, although technically it is. The reason is that hyperacuity is usually associated with stimuli that require resolving positional shifts, and not orientation. There is a log unit of difference between thresholds in these tasks.

P3.6. As a Suggestion: can the authors predict the accuracy of individual participants in single trials just by looking at the drift dynamics?

That’s a very interesting point that we indeed currently look at in another project. As a comment, we can add that by purely looking at the drift dynamics in the current data, we could not predict the accuracy (percent correct) of the participant. When comparing drift length or diffusion coefficients between trials with correct or false response, we do not observe a significant difference. Also, when adding an anatomical correlate and compare between trials where sampling density increases or decreases, there is no significant trend. We think that it is a more complex interplay between all the influencing factors that can perhaps be met by a model considering all drift dynamics, photoreceptor geometry and stimulus characteristics.

No changes were made to the manuscript.

**Recommendations for the authors:**

**Reviewing Editor (Recommendations For The Authors):**
As you will see, the reviewers were quite enthusiastic about your work, but have a few issues for your consideration. We hope that this is helpful. We'll consider any revisions in composing a final eLife assessment.
**Reviewer #1 (Recommendations For The Authors):**
R1.1: Discussion of myopia. Myopia takes a fair bit of space in the Discussion, but the paper does not include any subjects that are sufficiently myopic to test the predictions. I would suggest reducing the amount of space devoted to this issue, and instead making the prediction that myopia may help with resolution quickly. The introduction (lines 54-56) left me expecting a test of this hypothesis, and I think similarly that issue could be left out of the introduction.

We have removed this part from the Introduction and shortened the Discussion.

R1.2: Line 118: define CDC here.

Thank you for pointing this out, it is now defined at this location.

R1.3: Line 159-162: suggest breaking this sentence into two. This sentence also serves as a transition to the next section, but the wording suggests it is a result that is shown in the prior section. Suggest rewording to make the transition part clear. Maybe something like "Hence the spatial arrangement of cones only partially ... . Next we show that ocular motion and the associated ... are another important factor."

Text was changed as suggested.

R1.4.: Figure 3: The retina images are a bit hard to see - suggest making them larger to take an entire row. As a reader, I also was wondering about the temporal progression of the drift trajectories and the relation to the CDC. Since you get to that in Figure 4, you could clarify in the text that you are starting by analyzing distance traveled and will return to the issue of directed trajectories.

Visibility was probably an issue during the initial submission and review process where images were produced at lower resolution. The original figures are of sufficient resolution to fully appreciate the underlying cone mosaic and will later be able to zoom in the online publication.

We added a mention of the order of analysis in the Results section (LL 163-165)

R1.5: Line 176: define "sum of piecewise drift amplitude" (e.g. refer to Figure where it is defined).

We refer to this metric now as the drift length (as pointed out rightfully so by reviewer #2), and added its definition at this location.

R1.6: Lines 205-208: suggest clarifying this sentence is a transition to the next section. As for the earlier sentence mentioned above, this sounds like a result rather than a transition to an issue you will consider next.

This sentence was changed to make the transition clearer.

R1.7: Line 225: suggest starting a new paragraph here.

Done as suggested

**Reviewer #2 (Recommendations For The Authors):**
I don't have any major concerns, mostly suggestions and minor comments.R2.1: (1) The authors use piecewise amplitude as a measure of the amount of retinal motion introduced by ocular drift. However, to me, this sounds like what is normally referred to as the path length of a trace rather than its amplitude. I would suggest using the term length rather than amplitude, as amplitude is normally considered the distance between the starting and the ending point of a trace.

This was changed as suggested throughout the manuscript.

R2.2: (2) It would be useful to elaborate more on the difference between CDC and PCD, I know the authors do this in other publications, but to the naïve reader, it comes a bit as a surprise that drift directionality is toward the CDC but less so toward the PCD. Is the difference between these metrics simply related to the fact that defining the PCD location is more susceptible to errors, especially if image quality is not optimal? If indeed the PCD is the point of peak cone density, assuming no errors or variability in the estimation of this point, shouldn't we expect drift moving stimuli toward this point, as the CDC will be characterized by a slightly lower density? I.e., is the absence of a PCD directionality trend as strong as the trend seen for the CDC simply the result of variability and error in the estimate of the PCD or it is primarily due to the distribution of cone density not being symmetrical around the PCD?

Thank you for this comment. We already refer in the Methods section to the respective papers where this difference is analyzed in more detail, and shortly discuss it here.

To briefly answer the reviewer’s final question: PCD location is too variable, and ought to be avoided as a retinal landmark. While we believe there is value in reporting the PCD as a metric of maximum density, it has been shown recently (Reiniger et al., 2021; Warr et al., 2024; Wynne et al., 2022) and is visible in our own (partly unpublished) data, that its location will change with changing one or more of these factors: cone density metric, window size or cone quantity selected, cone annotation quality, image quality (e.g. across days), individual grader, annotation software, and likely more. Each of these factors alone can change the PCD location quite drastically, all while of course, the retina does not change. The CDC on the other hand, given its low-pass filtering nature, is immune to the aforementioned changes within a much wider range and will thus reflect the anatomical and, shown here, functional center of vision, better. However, there will always be individual eyes where PCD location and the CDC are close, and thus researchers might be inclined to also use the PCD as a landmark. We strongly advise against this. In a way, the PCD is a non-sense *location* while its dimension, *density*, can be a valuable metric, as density does not vary that much (see e.g. data on CDC density and PCD density reported in this manuscript).

Below we append a direct comparison of PCD vs CDC location stability when only one of the mentioned factors are changed. Sixteen retinas imaged on two different days were annotated and analyzed by the same grader with the same approach, and the difference in both locations are shown.

**Author response image 3. sa4fig3:** Reproducibility of CDC and PCD location in comparison. Two retinal mosaics which were recorded at two different timepoints, maximum 1 year apart from each other, were compared for 16 eyes. The retinal mosaics were carefully aligned. The retinal locations for CDC and PCD that were computed for the first timepoint were used as the spatial anchor (coordinate center), the locations plotted here as red circles (CDC) and gray diamonds (PCD) represent the deviations that were measured at the second timepoint for both metrics.

R2.3.: I don't see a statistical comparison between the drift angle tuning for CDC, PRL, and PCD. The distributions in Figure 4F look very similar and all with a relatively wide std. It would be useful to mark the mean of the distributions and report statistical tests. What are the data shown in this figure, single subjects, all subjects pooled together, average across subjects? Please specify in the caption.

We added a Rayleigh test to test each distribution for nun-uniformity and Kolmogorov-Smirnov tests to compare the distributions towards the different landmarks. We added the missing specifications to the figure caption of Figure 4 – figure supplement 1.

R2.4: I would suggest also calculating drift direction based on the average instantaneous drift velocity, similarly to what is done with amplitude. From Figure 3B it is clear that some drifts are more curved than others. For curved drifts with small amplitudes the start-point- end-point (SE) direction is not very meaningful and it is not a good representation of the overall directionality of the segment. Some drifts also seem to be monotonic and then change direction (eg. the last three examples from participant 10). In this case, the SE direction is likely quite different from the average instantaneous direction. I suspect that if direction is calculated this way it may show the trend of drifting toward the CDC more clearly.

In response to this and a comment of reviewer #1, we add a calculation of initial drift direction (and for increasing duration) and show it in Figure 4 – figure supplement 1. By doing so, we hope to capture initial directionality, irrespective of whether later parts in the path change direction. We find that directionality increases with increasing presentation duration.

R2.5: I find the discussion point on myopia a bit confusing. Considering that this is a rather tangential point and there are only two myopic participants, I would suggest either removing it from the discussion or explaining it more clearly.

We changed this section, also in response to comment R1.1.

R2.6: I would suggest adding to the discussion more elaboration on how these results may relate to acuity in normal conditions (in the presence of optical aberrations). For example, will this relationship between sampling cone density and visual acuity also hold natural viewing conditions?

We added only a half sentence to the first paragraph of the discussion. We are hesitant to extend this because there is very likely a non-straightforward relationship between acuity in normal and fully corrected conditions. We would predict that, if each eye were given the same type and magnitude of aberrations (similar to what we achieved by removing them), cone density will be the most prominent factor of acuity differences. Given that individual aberrations can vary substantially between eyes, this effect will be diluted, up to the point where aberrations will be the most important factor to acuity. As an example, under natural viewing conditions, pupil size will dominantly modulate the magnitude of aberrations.

R2.7: Line 398 - the point on the superdiffusive nature of drift comes out of the blue and it is unclear. What is it meant by "superdiffusive"?

We simply wanted to express that some drift properties seem to be adaptable while others aren’t. The text was changed at this location to remove this seemingly unmotivated term.

R2.8: Although it is true that drift has been assumed to be a random motion, there has been mounting evidence, especially in recent years, showing a degree of control and knowledge about ocular drift (eg. Poletti et al, 2015, JN; Lin et al, 2023, Current Biology).

We agree, of course. We mention this fact several times in the paper and adjusted some sentences to prevent misunderstandings. The mentioned papers are now cited in the Discussion.

R2.9: Reference 23 is out of context and should be removed as it deals with the control of fine spatial attention in the foveola rather than microsaccades or drift.

We removed this reference.

R2.10: Minor point: Figures appear to be low resolution in the pdf.

This seemed to have been an issue with the submission process. All figures will be available in high resolution in the final online version.

R2.11: Figure S3, it would be useful to mark the CDC at the center with a different color maybe shaded so it can be visible also on the plot on the left.

We changed the color and added a small amount of transparency to the PRL markers to make the CDC marker more visible.

R2.12: Figure S2, it would be useful to show the same graphs with respect to the PCD and PRL and maybe highlight the subjects who showed the largest (or smallest) distance between PRL and CDC.

Please find new Figure 4 supplement 1, which contains this information in the group histograms. Also, Figure 4 supplement 2 is now ordered by the distance PRL-CDC while the participant naming is kept as maximum acuity exhibited. In this way, it should be possible to infer the information of whether PRL-CDC distance plays a role. For us it does not seem to be crucial. Rather, stimulus onset and drift length were related, which is captured in Figure 4g.

R2.13: There is a typo in Line 410.

We could not find a typo in this line, nor in the ones above and below. “Interindividual” was written on purpose, maybe “intraindividual” was expected? No changes were made to the text.

References

Reiniger, J. L., Domdei, N., Holz, F. G., & Harmening, W. M. (2021). Human gaze is systematically offset from the center of cone topography. *Current Biology*, *31*(18), 4188–4193**.**
https://doi.org/10.1016/j.cub.2021.07.005

Ruderman, D. L., & Bialek, W. (1992). Seeing Beyond the Nyquist Limit. *Neural Computation*, *4*(5), 682–690. https://doi.org/10.1162/neco.1992.4.5.682

Warr, E., Grieshop, J., Cooper, R. F., & Carroll, J. (2024). The effect of sampling window size on topographical maps of foveal cone density. *Frontiers in Ophthalmology*, *4*, 1348950. https://doi.org/10.3389/fopht.2024.1348950

Williams, D. R. (1985). Aliasing in human foveal vision. *Vision Research*, *25*(2), 195–205. https://doi.org/10.1016/0042-6989(85)90113-0

Wynne, N., Cava, J. A., Gaffney, M., Heitkotter, H., Scheidt, A., Reiniger, J. L., Grieshop, J., Yang, K., Harmening, W. M., Cooper, R. F., & Carroll, J. (2022). Intergrader agreement of foveal cone topography measured using adaptive optics scanning light ophthalmoscopy. *Biomedical Optics Express*, *13*(8), 4445–4454. https://doi.org/10.1364/boe.460821